# Association of maternal heavy metal exposure during pregnancy with isolated cleft lip and palate in offspring: Japan Environment and Children's Study (JECS) cohort study

**Masato Takeuchi**[1]*, **Satomi Yoshida**[1], **Chihiro Kawakami**[2], **Koji Kawakami**[1], **Shuichi Ito**[2], **Japan Environment and Children's Study Group**[¶]

**1** Department of Pharmacoepidemiology, Graduate School of Medicine and Public Health, Kyoto University, Kyoto, Japan, **2** Graduate School of Medicine, Yokohama City University, Yokohama, Kanagawa, Japan

¶ Membership of the JECS Group is listed in the Acknowledgments.
* takeuchi.masato.3c@kyoto-u.ac.jp

## Abstract

### Background

Cleft lip and palate (cleft L/P) is one of the most common congenital anomalies and its etiology is assumed to be multifactorial. Recent epidemiological data involving a small number of participants suggested an association between perinatal exposure to heavy metals and cleft L/P in affected children. However, this association requires further investigation in a large cohort.

### Methods

This nested case–control study used a dataset of The Japan Environment and Children's Study, which is an ongoing research project to investigate the association between environmental factors and mother-child health. Participants were enrolled between 2011 and 2014. From the records of fetuses/children, we extracted data of isolated cleft L/P cases and matched children without cleft L/P at a ratio of 1:10. The exposures of interest were *in utero* exposure to four metals (mercury [Hg], lead [Pb], cadmium [Cd], and manganese [Mn]), which were sampled from mothers in the second/third trimester. Conditional logistic regression was used to assess the association between heavy metal exposure and isolated cleft L/P. Three sensitivity analyses were conducted to test the robustness of the findings, including the change in case definition and statistical methods.

### Results

Of 104,062 fetal records involving both live-birth and stillbirth, we identified 192 children with isolated cleft L/P and 1,920 matched controls. Overall, the blood metal levels were low (for example, median Pb level was 5.85, 6.22, and 5.75 μg/L in the total cohort, cases, and controls, respectively). Univariate and multivariate analyses showed that levels of none of the four heavy metals in the mother's blood during pregnancy were associated with the risk of

**Data Availability Statement:** Data are unsuitable for public deposition due to ethical restrictions and

legal framework of Japan. It is prohibited by the Act on the Protection of Personal Information (Act No. 57 of 30 May 2003, amendment on 9 September 2015) to publicly deposit the data containing personal information. Ethical Guidelines for Medical and Health Research Involving Human Subjects enforced by the Japan Ministry of Education, Culture, Sports, Science and Technology and the Ministry of Health, Labour and Welfare also restricts the open sharing of the epidemiologic data. All inquiries about access to data should be sent to: jecs-en@nies.go.jp. The person responsible for handling enquiries sent to this e-mail address is Dr Shoji F. Nakayama, JECS Programme Office, National Institute for Environmental Studies.

**Funding:** The JECS was funded by the Ministry of the Environment, Japan. The findings and conclusions of this article are solely the responsibility of the authors and do not represent the official views of the above government.

**Competing interests:** None.

cleft L/P in offspring; the adjusted odds ratios (per 1 μg/L increase) with 95% intervals were 0.96 (0.91–1.03), 1.01 (0.94–1.08), 1.00 (0.61–1.63) and 1.00 (0.97–1.03) for Hg, Pb, Cd and Mn, respectively. The results were consistent in all sensitivity analyses.

## Conclusions

Exposure to these four metals during pregnancy was not associated with isolated cleft L/P at the low exposure level in our cohort.

## Introduction

Cleft lip, cleft palate, and cleft lip and palate (hereafter collectively abbreviated as cleft L/P) are among the most common congenital defects worldwide [1]. The prevalence of cleft L/P has been reported as 1/700 live births [2], but regional differences in this prevalence have also been reported (1/500–1/2,500 live births) [3]. Japan is among the high-risk countries for cleft L/P, with a prevalence of 17.3–19.05/10,000 births [4, 5]. Cleft L/P is classified as syndromic and isolated (or equivocally non-syndromic) cases by etiology. The cause of syndromic cases is thought to be of genetic origin, whereas the cause for non-syndromic cases might be a combination of genetic and environmental factors [3, 6]. The established and possible environmental risk factors for isolated cleft L/P include maternal smoking, alcohol consumption, nutrition, viral infection, medicinal drugs, and occupational exposure to teratogens [2, 7].

In the past few years, several epidemiological studies have reported an association between maternal exposure to heavy metals and isolated cleft L/P in offspring. The metals suspected to increase the risk of cleft L/P include mercury (Hg), lead (Pb), cadmium (Cd), arsenic, and nickel [8–12]. The hazard of these metals on child health is substantial, not limited to the potential risk for cleft L/P [13–16]. Despite the small cohort sizes, those studies suggested that heavy metal exposure *in utero* could be an unrecognized risk factor for developing cleft L/P. At the population level, the source of metal levels has now shifted from the atmosphere to foods in Japan as well as many countries [17]. Because such a metal exposure from contaminated food is potentially modifiable by monitoring or education, this association merits further investigation in a large cohort.

The Japan Environment and Children's Study (JECS) is an ongoing research project that has collected data from 100,000 mother-child pairs in a birth cohort [18]. Using the JECS dataset, we aimed to investigate whether maternal exposure to heavy metals was associated with the risk of isolated cleft L/P in offspring.

## Methods

### Overview of the JECS

The JECS, which is ongoing in multiple Regional Centers in Japan, is a nationwide cohort study that was originally organized to investigate the association between environmental factors and mother-child health from pregnancy to 13 years of age [18, 19]. The study protocol was approved by the Institutional Review Board of the Ministry of the Environment and relevant Regional Centers. The recruitment period for the study was January 2011 to March 2014. The JECS aimed to register approximately 100,000 pregnant women and their offspring. This cohort profile was similar to that of the vital statistics in Japan [20]. To recruit participants, pregnant women were invited at the nearest Co-operating health care providers and/or local

government offices that issue maternal and child health handbooks. A written informed consent was obtained from all participants.

The visit schedule of pregnant women and their children has been described elsewhere [18]. Our study used data that were extracted from questionnaires answered by the mother at the first trimester (MT1) and the second/third trimester (MT2), maternal blood sample measurements of the mother at MT1 and MT2, and medical record transcripts of the newborn at birth (Dr0m) and 1 month (Dr1m).

## Definition of outcome

The present study used the dataset jecs-ag-20160424, which was released in June 2016 and revised in October 2016, along with the supplementary dataset jecs-ag-20160424-sp1. This dataset contained information on 39 major congenital anomalies (S1 Table) that were recognized at birth (Dr0m) and at the 1-month checkup (Dr1m).

Information relevant to congenital anomalies was extracted from medical records at each Co-operating health care provider, and was further validated by pediatricians at the Medical Support Centre. The details of this process and the descriptive results were previously published [21]. Unlike this previous study, we used the information of children whose birth outcome was miscarriage or stillbirth in order to attempt to minimize "live birth" bias [22].

The primary outcome of interest was isolated cleft L/P. We defined the cohort of isolated cleft L/P by the following both conditions: 1) children with a diagnosis of cleft lip, cleft palate, or cleft lip and palate in either Dr0m or Dr1m; and 2) children without other congenital malformation(s) recorded at either Dr0m or Dr1m. In the primary analysis, the control cohort comprised children without cleft L/P and other congenital anomalies, to be comparable with our definition of isolated cases.

## Exposure

The primary exposures of interest were levels of four selected heavy metals measured at MT2 (Hg, Pb, Cd, and manganese [Mn]). These metals were selected because of concerns for a negative health consequence of the mother and child [13–16] as well as the research interest of JECS topic groups, and the details of measurement methods, quality control, and unit conversion were published elsewhere [23]. In this previous publication, blood samples from 20,000 of the 100,000 women were selected at random, measured, and reported. We obtained the data from all 100,000 samples and confirmed that metal levels were similarly distributed as previously reported [23] (S2 Table).

If heavy metal measurements were not available for the mother, the child was excluded from our study cohort.

## Statistical analysis

To compare the characteristics of participants, we used the Wilcoxon rank sum test (as most variables had a skewed distribution) and the chi-square test (or Fisher's exact test where appropriate) for continuous and categorical variables, respectively.

The primary analysis was conducted by nested case–control design [24]. In a case–control study, four to five controls are often selected per case, but whether this is the best practice for all datasets is unclear [25]. We selected up to 10 controls per case because the JECS study collected data from 100,000 mother-child pairs and occurrence of the primary outcome was expected to be infrequent.

The optimal matching procedure was selected because this method attempts to minimize the global distance between matched datasets and empirically offers the best balance within

each matched pair [26, 27]. The distance was calculated with the Mahalanobis distance. Covariates for matching were maternal age, psychological stress measured by the K6 score (MT1) [28], gestational weeks of blood sampling at MT2, folic acid intake estimated from a food-frequency questionnaire (MT1) [29], alcohol intake (self-reported; MT1), smoking (self-reported; MT1), education level (MT2), body mass index before pregnancy (MT1), diabetes before pregnancy, intake of supplements (self-reported; MT1), and Regional Center. These covariates were selected *a priori* from a literature review [2, 3, 7, 30] and to mitigate the influence of unmeasured factors clustered in the same region (Regional Center) [31]. For missing variables, categorical data were treated as they were (i.e., creating a category of "NA" referred to as the missing indicator approach), whereas individuals with missing continuous data were excluded from the primary analysis.

For the matched dataset, we performed univariate and multivariate conditional logistic regression. In the multivariate model, the sex of the child and the concentrations of four metals were simultaneously entered. We found that metal concentrations were heavily skewed owing to outliers. We performed the same analysis using log-transformed data, but the results were unchanged (S3 Table). Therefore, we present non-transformed data for interpretability [32].

All statistical analyses were conducted using R statistical software, version 4.03 (https://cran.r-project.org/). A *p* value <0.05 indicated statistical significance.

## Secondary analyses

We planned three sensitivity analyses. The first analysis applied multiple imputations for missing variables. In this analysis, we assumed that the missing mechanism was "missing completely at random" and missing variables could be predicted by other covariates prepared for matching [33]. Imputations were repeated 20 times. The second sensitivity analysis used all datasets of cleft L/P. In this analysis, children who were eligible for matching were allowed to have congenital anomalies other than cleft L/P. The matching was repeated in the above-mentioned sensitivity analyses and multivariate conditional logistic regression was also conducted. In addition, we planned the sensitivity analysis to be limited to live births. However, all cleft L/P cases were live-born; as a result, we omitted this analysis.

One post-hoc sensitivity analysis was added. Random-effect logistic regression was performed for isolated cleft L/P cases and all children without cleft L/P and a congenital anomaly to examine whether our results were susceptible to "overmatching" (discussed below). The covariates of this model were those used for the matching (except for Regional Center) and heavy metal concentrations, and a Regional Center identifier was entered into the model as a random intercept.

## Results

### Primary cohort

Of 104,062 fetal records with parental consent in the dataset jecs-ag-20160424, 10,451 were removed owing to missing data (Fig 1). Of 95,092 remaining children, 226 children had cleft L/P (23.7/10,000 children) and 192 cases fulfilled our definition of isolated cleft L/P. A total of 92,796 children who did not have cleft L/P and the major congenital anomalies were eligible for matching controls (Table 1); there were 172 records of stillbirths, all of which were reported in the control pool. More details on the metal concentrations measured, including mean value and range, are shown in S4 Table. Among the 192 isolated cases of L/P, cleft lip, cleft palate, and cleft lip and palate were described in 67, 44, and 91 cases, respectively (a subset of the children had different categories in Dr0m and Dr1m, and therefore, the total number exceeded 192).

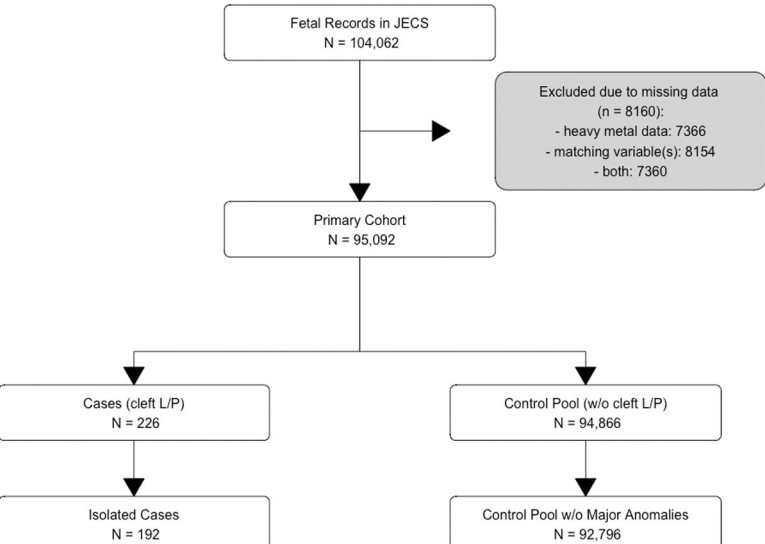

**Fig 1. Flow diagram of study cohort.** Note: Only matching variables of continuous data were excluded (e.g., maternal age).

The matched dataset involved 192 cases and 1,920 controls (Table 1). Cases and selected controls were well-matched overall regarding matching variables and the major birth outcomes. Univariate and multivariate analyses showed that maternal blood levels of none of the heavy metals were associated with the risk of cleft L/P in the offspring (Table 2).

## Secondary analyses

Three sensitivity analyses—two for isolated cases and one involving syndromic cases—were conducted (Table 3). First, by matching with multiple imputations (using the data of 94,567 children with information on heavy metals in maternal blood samples), we created a matched cohort that comprised 192 cases of isolated cleft L/P and 1,920 controls without major anomalies. In these imputed datasets, we found no association between mother's blood heavy metal levels and the risk of cleft L/P in children. Second, random-effect logistic regression without matching was performed using data of 92,988 children who had complete data (questionnaire and maternal blood levels of heavy metals). The results were similar to those of the primary analysis, with a non-significant effect of heavy metals on the risk of isolated cleft L/P (S5 Table). The third sensitivity analysis was expanded to all cases of cleft L/P (i.e., also including syndromic cases). The repeated matching created a pairing of 226 cases and 2,260 controls. As in the above analyses, the risk of cleft L/P was unrelated to the mother's heavy metal levels during pregnancy.

## Discussion

This study examined the association between exposure to Hg, Pb, Cd, and Mn *in utero* and isolated cleft L/P in offspring using a nationwide, large-scale birth cohort in Japan. Among matched pairs of 192 cases and 1,920 controls, the risk of cleft L/P was not associated with maternal heavy metal levels during pregnancy at a relatively low exposure. This result was confirmed by three sensitivity analyses, including changes in case definition and analytical methods.

**Table 1. Characteristics of the primary cohort and matched pairs[1].**

| | Primary cohort (n = 95,092) | Isolated cleft L/P (n = 192) | Control (n = 1920) |
|---|---|---|---|
| **Age** (mother: yrs) | 31 (28–35) | 31 (27–34) | 31 (28–34) |
| **GA** (weeks) | 39 (38–40) | 39 (38–40) | 39 (38–40) |
| **Birth weight** (g) | 3020 (2770–3280) | 2985 (2659–3268) | 3020 (2776–3280) |
| **Live birth** | 94,913 (99.8%) | 192 (100%) | 1916 (99.8%) |
| **Male sex** | 48,708 (51.2%) | 111 (57.8%) | 967 (50.4%) |
| **Psychological stress** | Yes: 3273 (3.4%) | Yes: 8 (4.1%) | Yes: 80 (4.1%) |
| | No: 91447 (96.2%) | No: 181 (94.2%) | No: 1810 (94.2%) |
| | NA: 372 (0.39%) | NA: 3 (1.5%) | NA: 30 (1.5%) |
| **FA intake** (ug/day) | 246 (178–338) | 256.5 (170–378) | 242 (182–331) |
| **Alcohol** | Never-drinker: 32,741 (34.4%) | Never-drinker: 75 (39.1%) | Never-drinker: 728 (37.9%) |
| | Ex-drinker: 52,591 (55.3%) | Ex-drinker: 103 (53.6%) | Ex-drinker: 1062 (55.3%) |
| | Current-drinker: 9347 (9.8%) | Current-drinker: 13 (6.8%) | Current-drinker: 120 (6.3%) |
| | NA: 413 (0.43%) | NA: 1 (0.52%) | NA: 10 (0.52%) |
| **Smoking** | Never-smoker:54,923 (57.8%) | Never-smoker: 108 (56.3%) | Never-smoker: 1094 (57.0%) |
| | Ex-smoker: 22,398 (23.6%) | Ex-smoker: 50 (26.0%) | Ex-smoker: 492 (25.6%) |
| | Quit after pregnancy: | Quit after pregnancy: | Quit after pregnancy: |
| | 12,537 (13.2%) | 27 (14.1%) | 264 (13.8%) |
| | Current smoker 4546 (4.8%) | Current smoker 7 (3.6%) | Current smoker: 70 (3.6%) |
| | NA: 688 (0.72%) | NA: 0 (0%) | NA: 0 (0%) |
| **Education** | High school: 34,231 (36.0%) | High school: 78 (40.6%) | High school: 752 (39.2%) |
| | College: 39,570 (41.6%) | College: 69 (35.9%) | College: 712 (37.1%) |
| | University or higher: | University or higher: | University or higher: |
| | 20,116 (21.2%) | 41 (21.4%) | 416 (21.7%) |
| | NA: 1175 (1.2%) | NA: 4 (2.1%) | NA: 40 (2.1%) |
| **BMI** (kg/m$^2$) | 20 (19–22) | 20 (19–23) | 20 (19–22) |
| **Pre-pregnancy DM** | Yes: 2 (<0.1%) | Yes: 0 (0%) | Yes: 0 (0%) |
| **Supplement intake** | Yes: 10,260 (10.8%) | Yes: 31 (16.1%) | Yes: 300 (15.7%) |
| | No: 76,429 (80.4%) | No: 146 (76.0%) | No: 1457 (75.9%) |
| | NA: 8403 (8.8%) | NA: 16 (8.3%) | NA: 163 (8.5%) |
| **GA of exam** (weeks) | 27 (25–29) | 27 (25–29.5) | 27 (25–29) |
| **Heavy metal** (µg/L) | | | |
| Hg | 3.63 (2.54–5.19) | 3.64 (2.60–4.98) | 3.54 (2.52–5.27) |
| Pb | 5.85 (4.70–7.33) | 5.84 (4.49–7.18) | 5.75 (4.69–7.14) |
| Cd | 0.66 (0.50–0.90) | 0.66 (0.49–0.90) | 0.66 (0.49–0.89) |
| Mn | 15.4 (12.6–18.7) | 15.2 (12.5–18.5) | 15.4 (12.7–18.5) |

[1]For continuous variables, the median with interquartile range was reported.

GA, gestational age; FA, folic acid; BMI, body mass index; DM, diabetes mellitus; Hg, mercury; Pb, lead; Cd, cadmium; Mn, manganese; NA, not available.

**Table 2. Association between heavy metal concentrations and cleft L/P.**

| | Univariate | p-value | Multivariate[1] | p-value |
|---|---|---|---|---|
| **Hg[2]** | OR: 0.97 (95% CI: 0.91–1.03) | 0.29 | OR: 0.96 (95% CI: 0.91–1.03) | 0.26 |
| **Pb[2]** | OR: 1.01 (95% CI: 0.94–1.08) | 0.81 | OR: 1.01 (95% CI: 0.94–1.08) | 0.78 |
| **Cd[2]** | OR: 0.99 (95% CI: 0.62–1.56) | 0.95 | OR: 1.00 (95% CI: 0.61–1.63) | 0.99 |
| **Mn[2]** | OR: 1.00 (95% CI: 0.97–1.03) | 0.98 | OR: 1.00 (95% CI: 0.97–1.03) | 0.96 |

[1]Adjusted for sex and concentrations of the four metals.

[2]Per 1 µg/L increase.

OR, odds ratio; CI, confidence interval.

**Table 3. Sensitivity analyses.**

| | Multivariate model[1] (OR with 95% CI) | p-value |
|---|---|---|
| **1: MI for missing data** (cases: 192, controls: 1920) | | |
| Hg[2] | 0.96 (0.89–1.02) | 0.17 |
| Pb[2] | 0.98 (0.93–1.04) | 0.50 |
| Cd[2] | 0.96 (0.62–1.48) | 0.84 |
| Mn[2] | 1.00 (0.96–1.03) | 0.87 |
| **2: Random-effects model[3]** (cases: 187, comparators: 91,164) | | |
| Hg[2] | 0.97 (0.92–1.04) | 0.42 |
| Pb[2] | 0.99 (0.93–1.04) | 0.64 |
| Cd[2] | 0.95 (0.61–1.46) | 0.81 |
| Mn[2] | 1.00 (0.96–1.03) | 0.87 |
| **3: Including both isolated and syndromic cases** (cases:218, controls: 2180) | | |
| Hg[2] | 0.96 (0.90–1.02) | 0.18 |
| Pb[2] | 1.01 (0.95–1.07) | 0.74 |
| Cd[2] | 0.88 (0.56–1.39) | 0.58 |
| Mn[2] | 0.99 (0.96–1.03) | 0.75 |

[1]Adjusted for sex and concentrations of the four metals for analyses 1 and 3, and adjusted for 10 matching covariates, sex, and concentrations of the four metals for analysis 2.

[2]Per 1 µg/L increase.

[3]Without matching.

OR, odds ratio; CI, confidence interval; MI, multiple imputations.

Several epidemiological studies in recent years and experimental investigations using animal models have reported an association between heavy metal exposure and cleft L/P. For example, Pi et al. reported that higher Hg and Cd concentrations in placental tissues were associated with a higher risk of neonatal orofacial clefts in a dose-dependent manner [9]. Similarly, Ni et al. found associations between higher Cd, Pb, arsenic, and nickel concentrations in umbilical cord tissue and a risk of orofacial clefts in offspring [10]. Additionally, maternal occupational Cd exposure and fish consumption with supposed contamination by heavy metals are also reported as risk factors for cleft L/P [11, 12]. These findings were partly supported by experiments in rodents [8, 34, 35]. These previous reports suggest that maternal heavy metal exposure is an unrecognized risk factor for cleft L/P in children and this issue merits further investigation.

However, in contrast to these reports, our study did not find an association between heavy metal concentrations in pregnant women and cleft L/P in children. There are several possible explanations for this discrepancy between our study and others. First, the level of exposure was low overall in the JECS cohort. Previous JECS research showed that a minority of samples (none to <1%) exceeded the "action level" of Hg, Pb, and Cd—the exposure level associated with an adverse health effect [23]. If the risk threshold for cleft L/P is at a higher level, we could have missed such an association. Unfortunately, direct comparison of exposure levels with data from previous studies is difficult for reasons such as differences in tissue samples (e.g., maternal blood vs. placental tissue) or in the timing of exposure assessment. A second explanation is that very early fetal losses (e.g., before perception of pregnancy) were not included in the JECS dataset. This might have led to two types of selection bias [36] as follows: bias resulting in 1) a different distribution of heavy metal exposure between the analyzed dataset and the

entire pregnant population, and 2) bias resulting in a reduced number of fetuses susceptible to cleft L/P (with or without other malformations) in mothers with a high heavy metal exposure. Although this could explain the negative results of our study, it might not address the discrepancy with previous studies because this bias was also inherent to other epidemiological studies. Third, we applied a nested case–control analysis, and this study design could have led to overmatching bias. If matching is undertaken for intermediate variables and/or variables associated with the exposure, but not with a risk of disease, the crude odds ratio would be closer to 1 [37, 38]. Although matching variables in this study were selected through a literature review, the pathogenesis of cleft L/P has not been fully determined. To examine whether the matching procedure negated a possible association, we performed a post-hoc sensitivity analysis for an unmatched dataset using multivariate logistic regression. We found that the results were similar for the unmatched data to those for the matched dataset. Therefore, we speculate that the potential for overmatching was minimal.

## Strengths and limitations of the study

The JECS has collected more than100,000 records of mother-infant pairs. This large-scale cohort enabled us to construct well-matched pairs of cases and controls. This may eventually lead to a relatively narrow confidence band on the risk scale of cleft L/P (Table 2). However, this study has several limitations. First, blood samples were collected only once, during the second/third trimester, which is beyond the critical period of organogenesis [39, 40], because of a constraint of project resources. The major source of the four metals examined (Hg, Pb, Cd, and Mn) was thought be food or smoking (Cd only) [23], and eating or smoking patterns could change between the early stage and mid-pregnancy. However, fetal development of lip and palate starts as early as 4–5 weeks of gestation [2], the period around or before pregnancy perception in most women. Blood sample collection prior to this period is a challenge in a large-scale cohort; it should be acknowledged that there is often a trade-off between research feasibility and scientific basis. Second, embryological development of the lip and palate differs [40]. Therefore, cleft lip, cleft palate, and cleft lip and palate should be ideally analyzed as three distinct entities. However, there were inconstancies in the classification between the Dr0m and Dr1m records in a subset of children with cleft L/P. Both records were extracted from medical record transcripts, and documents submitted to the Program Office were reviewed and confirmed by two independent pediatricians [21]. Accordingly, the authors could not reclassify children with "inconsistent" diagnoses, and decided to analyze them as one group of cleft L/P. Third, misclassification between isolated and syndromic cases was possible. We identified isolated cases with cleft L/P without other major anomalies (S1 Table). However, syndromic cases with multiple minor anomalies or even major anomalies recognized later in life could not be specified by our approach. Furthermore, there appears to be no universally accepted definition to constitute syndromic cases [7], resulting in the potential for misclassification. However, a similar case definition was used to select controls without cleft L/P. Therefore, the effect of misclassification on the risk estimate of cleft L/P was likely offset in our study. Additionally, we conducted a sensitivity analysis involving all cleft L/P cases, and confirmed no association between heavy metal exposure and the risk of cleft L/P also in this population. Fourth, our primary cohort included stillbirths. However, although there were 172 cases of stillbirth in the control pool, cases of cleft L/P were eventually all live-born. As a result, the effect of live birth bias could not be evaluated. Finally, submucous cleft palate may be underdiagnosed at birth or at 1 month of age; according to one case series, submucous cleft palate was first diagnosed at the mean age of 4.9 years [41]. However, the prevalence of cleft L/P in the JECS cohort was equivalent to or even higher than that previously reported in Japan

[4, 5]. The reason for this finding might be because underreporting was minimized by prospective data collection and structured quality control of data in the JECS [21]. Although we were not able to determine whether children with submucous cleft palate were included in the JECS dataset, we assume that there was minimal underdiagnosis/underreporting.

In summary, we did not observe an association between maternal exposure to Hg, Pb, Cd, or Mn and the risk of isolated cleft L/P in offspring in a large-scale birth cohort in Japan. However, this finding should be interpreted with caution because the exposure levels in our cohort were relatively low. Future research may be warranted in areas where higher exposure to heavy metals is anticipated. Additionally, as a future research implication, measurement of exposure at the early phase of pregnancy or its surrogate marker may be crucial to accurately determine the association of maternal heavy metal exposure and congenital malformations in offspring, including cleft L/P.

## Supporting information

**S1 Table. Major congenital anomalies recorded in Dr0m and Dr1m.**
(DOCX)

**S2 Table. Heavy metal concentrations.**
(DOCX)

**S3 Table. Multivariate analysis with log-scale metal concentrations.**
(DOCX)

**S4 Table. Detailed distribution data of heavy metal concentrations.**
(DOCX)

**S5 Table. Results from random-effect multivariate analysis (sensitivity analysis 2).**
(DOCX)

## Acknowledgments

We are grateful to all of the participants of the JECS and to all individuals involved in data collection. We thank Ellen Knapp, PhD, from Edanz Group (https://en-author-services.edanz.com) for editing a draft of this manuscript.

Members of the JECS Group as of 2021: Michihiro Kamijima (principal investigator, Nagoya City University, Nagoya, Japan), Shin Yamazaki (National Institute for Environmental Studies, Tsukuba, Japan), Yukihiro Ohya (National Center for Child Health and Development, Tokyo, Japan), Reiko Kishi (Hokkaido University, Sapporo, Japan), Nobuo Yaegashi (Tohoku University, Sendai, Japan), Koichi Hashimoto (Fukushima Medical University, Fukushima, Japan), Chisato Mori (Chiba University, Chiba, Japan), Shuichi Ito (Yokohama City University, Yokohama, Japan), Zentaro Yamagata (University of Yamanashi, Chuo, Japan), Hidekuni Inadera (University of Toyama, Toyama, Japan), Takeo Nakayama (Kyoto University, Kyoto, Japan), Hiroyasu Iso (Osaka University, Suita, Japan), Masayuki Shima (Hyogo College of Medicine, Nishinomiya, Japan), Youichi Kurozawa (Tottori University, Yonago, Japan), Narufumi Suganuma (Kochi University, Nankoku, Japan), Koichi Kusuhara (University of Occupational and Environmental Health, Kitakyushu, Japan), and Takahiko Katoh (Kumamoto University, Kumamoto, Japan).

## Author Contributions

**Conceptualization:** Koji Kawakami, Shuichi Ito.

**Data curation:** Chihiro Kawakami.

**Formal analysis:** Masato Takeuchi.

**Methodology:** Masato Takeuchi.

**Project administration:** Shuichi Ito.

**Resources:** Koji Kawakami.

**Supervision:** Shuichi Ito.

**Visualization:** Masato Takeuchi.

**Writing – original draft:** Masato Takeuchi.

**Writing – review & editing:** Satomi Yoshida.

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
