## [Decision Letter · Decision Letter 0]

24 Nov 2021

PONE-D-21-29327Association of maternal heavy metal exposure during pregnancy with isolated cleft lip and palate in offspring: Japan Environment and Children’s Study (JECS) cohort studyPLOS ONE

Dear Dr. Takeuchi,

Thank you for submitting your manuscript to PLOS ONE. After careful consideration, we feel that it has merit but does not fully meet PLOS ONE’s publication criteria as it currently stands. Therefore, we invite you to submit a revised version of the manuscript that addresses the points raised during the review process.

We look forward to receiving your revised manuscript.

Kind regards,

Forough Mortazavi

Academic Editor

PLOS ONE

Journal Requirements:

Additional Editor Comments (if provided):

Dear authors,

Thank you for submitting your manuscript to PLOS ONE. Please consider the reviewers’ comments carefully and revise the manuscript accordingly. As the reviewers stated, manganese and selenium are not considered as heavy metals. The justification for considering manganese and selenium as heavy metals is not sufficient and convincing since both of these are among the prescribed minerals during pregnancy. You need to cite at least one more study with regard to this matter in the introduction section in addition to the reference 18. Also, arsenic is not included in your list of five heavy metals without any justification. PLS explain the scientific evidence and rationale for the selection of the five heavy metals. PLS provide the statistics regarding the rate of exposure to heavy metals as well as the rate of cleft lip/palate in Japan in the introduction section to justify conducting this study.

This study did not find any relationship between exposure to heavy metals and cleft lip/palate in Japan; therefore, it is not necessary to measure the effects of other risk factors of cleft lip/palate because they would not change the relationship between exposure to heavy metals and anomalies. But if you have collected data regarding other risk factors, I suggest that in addition to heavy metals, you consider other risk factors in relation to cleft lip/palate in your analysis.

Reviewers' comments:

Reviewer's Responses to Questions

**Comments to the Author**

1. Is the manuscript technically sound, and do the data support the conclusions?

Reviewer #1: Partly

Reviewer #2: Yes

2. Has the statistical analysis been performed appropriately and rigorously? 

Reviewer #1: No

Reviewer #2: No

3. Have the authors made all data underlying the findings in their manuscript fully available?

Reviewer #1: No

Reviewer #2: Yes

4. Is the manuscript presented in an intelligible fashion and written in standard English?

Reviewer #1: Yes

Reviewer #2: Yes

5. Review Comments to the Author

Reviewer #1: Although the authors pick an important topic on heavy metal exposure and CL/P in Japan, no significant association was found, and there are many reasons, including over matched the controls (1:10), low blood metal levels, although the authors only report the lead level in the abstract, statistical analyses and et al. There are some major concerns that the authors should pay attention, which was not limited to the following:

1. Selenium did not belong to heavy metals, and the correlation of selenium with birth defects was not coincided with other heavy metals which the authors needs to be aware of it. Line 130-131: please check it.

2. Line 75-78: the risk factors was too general and it did not mentioned the heavy metals, i.e., lack of introduction of why did the author conduct the current research?

3. Line 83: needs to clarify that “Because exposure to heavy metals is potentially modifiable”

4. Controversies about the description: Line 117-118 vs Line 219-221 vs Line 222-224.

Line 117-118: Unlike this previous study, we used information of children whose birth outcome was miscarriage or stillbirth, with the aim of minimizing “live birth” bias (17).

Line 219-221: Fourth, our primary cohort included still births. However, cases of cleft L/P were eventually all live born. As a result, the effect of live birth bias could not be evaluated”.

Line 222-224: “A second explanation is that early fetal losses (e.g., before perception of pregnancy) were not included in the JECS dataset”

5. Statistical problem:

(1) Mean metal level, median metal level and Median with interquartile range are both used in the study, due to the distribution of metal concentration, the expression needs to be uniformed.

(2) Line 165-166: We found that metal concentrations were heavily skewed owing to outliers. How to tackle with “outliers”?

(3) The control variables were not enough in the multivariate analysis, which only sex and other metals controlled, how about other risk factors related to CL/P?

(4) As there are several metals in the analysis, proper statistical analysis may need, for example, BKMR analysis may be considered.

6. It’s hard to understand the notes under S1 Table: Major congenital anomalies recorded in Dr0m and Dr1m1 (1: Cleft lip, cleft palate, and cleft lip and palate are listed, bur are omitted from this table.)

Reviewer #2: The manuscript highlighted the associations of prenatal exposures of heavy metals and selected congenital anomalies. The manuscript is well written. The study design, analysis, results, and interpretation are acceptable. I have the following observation, which needs to be addressed before acceptance of the articles.

The introduction needs further elaboration about the burden of metal exposures worldwide. Further, the justification of including manganese and selenium is not sufficient. The authors should specify the health impact of metals on the outcome of interest, including the risk estimates.

It is not clear why other cases of congenital anomalies were not considered in the analysis.

Did the author graphically observe the associations by scatter plots (with Loess)? Linear associations are not always observed between toxic exposures and health outcomes. Whether there is any effect in the low exposure levels should be checked graphically. Furthermore, the authors should also explore the associations by grouping the exposures into tertile/quartile.

One of the weaknesses of the study is that exposure at very early gestation is not available. This weakness should be discussed. Further, are there any other unmeasured metals (such as arsenic) that may influence the observed association?

It is better if the authors add a separate table for exposures, including the mean, minimum and maximum values.

It is better for a reader if the authors add paragraphs on the strengths and limitations of the study.

Minor issues: Page 23, Line 220: missing full-stop; Page 15, Table 1: replace ‘media’ with ‘median’ at the end.

6. PLOS authors have the option to publish the peer review history of their article (what does this mean?). If published, this will include your full peer review and any attached files.

Reviewer #1: **Yes: **Jufen Liu

Reviewer #2: **Yes: **Anisur Rahman

---

## [Author Response · Author response to Decision Letter 0]

5 Jan 2022

Editors’ comments

Comment E1 

Please include additional information regarding the survey or questionnaire used in the study and ensure that you have provided sufficient details that others could replicate the analyses. For instance, if you developed a questionnaire as part of this study and it is not under a copyright more restrictive than CC-BY, please include a copy, in both the original language and English, as Supporting Information.

Response to comment E1

The detailed items of questionnaire, including the responses, cannot be shared because of JECS regulations.

Comment E2

We note that you have included the phrase “data not shown” in your manuscript. Unfortunately, this does not meet our data sharing requirements. PLOS does not permit references to inaccessible data. We require that authors provide all relevant data within the paper, Supporting Information files, or in an acceptable, public repository. Please add a citation to support this phrase or upload the data that corresponds with these findings to a stable repository (such as Figshare or Dryad) and provide and URLs, DOIs, or accession numbers that may be used to access these data. Or, if the data are not a core part of the research being presented in your study, we ask that you remove the phrase that refers to these data.

Response to comment E2

Data formerly described as “not shown” has been added as supplemental information (S3 Table). 

Comment E3

As the reviewers stated, manganese and selenium are not considered as heavy metals. The justification for considering manganese and selenium as heavy metals is not sufficient and convincing since both of these are among the prescribed minerals during pregnancy. You need to cite at least one more study with regard to this matter in the introduction section in addition to the reference 18. Also, arsenic is not included in your list of five heavy metals without any justification. Please explain the scientific evidence and rationale for the selection of the five heavy metals. Please provide the statistics regarding the rate of exposure to heavy metals as well as the rate of cleft lip/palate in Japan in the introduction section to justify conducting this study.

Response to comment E3

We thank the Editor as well as the Reviewers for pointing out this important issue. After discussion among the authors, we removed selenium from the current analysis. The analyses were updated in this revision accordingly, and we found that the results were mostly unchanged. By contrast, we retained manganese because this metal can be broadly classified as a heavy metal (density: 7.2 g/cm3) and high concentrations of manganese may be associated with congenital anomalies (Crit Rev Oncol Hematol. 2002;42(1):25–34). 

As for arsenic, it is now being measured and data will be available in the future (personal communication with Dr. Nakayama, the author of Ref. 23). According to Dr. Nakayama, arsenic cannot be assessed in tandem with the other metals examined, given the situation in Japan. Residents of Japan ingest organic arsenic (assumed to be less toxic than inorganic arsenic) from seaweed, whereas exposure to inorganic arsenic is thought to be low. Accordingly, the exposure level of organic and inorganic arsenic should be specified separately to evaluate the environmental hazard of each. However, the method of measuring arsenic differs from the measurement of other metals. This was the reason for data on arsenic being unavailable to date. Again, the measurement of arsenic is now ongoing. There was another operational reason for selecting five metals. JECS has several topic groups, such as congenital anomaly or child development, and each group has a variety of research interests. In addition, JECS measured numerous environmental factors other than metals (e.g., household chemicals and allergens). From the initial research themes submitted to the JECS core team, the priority was set for the five metals selected; as noted above, data on other metals, such as copper and zinc, would be subsequently updated. At first, JECS measured the levels of five metals in approximately 20% of samples, then expanded the measurement to the entire cohort. This process required several years, illustrating how time-consuming the analysis of 100,000 blood samples was. 

 Finally, although the Editor requested adding the statistics regarding the rate of exposure to heavy metals to the Introduction, little is known about the overall exposure level to metals in Japan, a situation that partly prompted the JECS. In response to the Reviewer concern, we have instead added a concise description of global metal exposure and possible adverse health effects to improve the flow of the Introduction.

Comment E4

This study did not find any relationship between exposure to heavy metals and cleft lip/palate in Japan; therefore, it is not necessary to measure the effects of other risk factors of cleft lip/palate because they would not change the relationship between exposure to heavy metals and anomalies. But if you have collected data regarding other risk factors, I suggest that in addition to heavy metals, you consider other risk factors in relation to cleft lip/palate in your analysis.

Response to comment E4

We thank the Editors for this comment. We performed a multivariable logistic regression in the sensitivity analysis, and list the odds ratios for other risks obtained from this analysis in S5 Table.

Reviewers' comments:

Reviewer #1

Comment 1-1

Selenium did not belong to heavy metals, and the correlation of selenium with birth defects was not coincided with other heavy metals which the authors needs to be aware of it. Line 130-131: please check it.

Response to Comment 1-1

We thank the Reviewer for this comment. As we responded to the Editor’s comment above (Comment E3), selenium has been removed from the revised version.

Comment 1-2

Line 75-78: the risk factors was too general and it did not mentioned the heavy metals, i.e., lack of introduction of why did the author conduct the current research?

Response to Comment 1-2

We thank the Reviewer for commenting this important point; a similar comment was made by the Editor and another Reviewer. To explain the rationale for this study, we added a brief description of the hazards of metals on child health.

Comment 1-3

Line 83: needs to clarify that “Because exposure to heavy metals is potentially modifiable”

Response to Comment 1-3

We have added the explanation as follows:

“At the population level, the source of metal levels has now shifted from the atmosphere to foods in Japan as well as many countries (17). Because such a metal exposure from contaminated food is potentially modifiable by monitoring or education, this association merits further investigation in a large cohort.”

Comment 1-4

Controversies about the description: Line 117-118 vs Line 219-221 vs Line 222-224.

Line 117-118: Unlike this previous study, we used information of children whose birth outcome was miscarriage or stillbirth, with the aim of minimizing “live birth” bias (17).

Line 219-221: Fourth, our primary cohort included still births. However, cases of cleft L/P were eventually all live born. As a result, the effect of live birth bias could not be evaluated”.

Line 222-224: “A second explanation is that early fetal losses (e.g., before perception of pregnancy) were not included in the JECS dataset”

Response to Comment 1-4

We thank the Reviewer for pointing out inconsistent descriptions. The sentences have been modified as follows:

Line 117-118: Unlike this previous study, we used the information of children whose birth outcome was miscarriage or stillbirth in order to attempt minimizing “live birth” bias (17).

(To the Reviewer: Please note that we explained the rationale for including miscarriage or stillbirth in the planning phase, in contrast to the cohort profile paper of JECS; the JECS core team is very sensitive to even minor differences in the case numbers between studies.)

Line 219-221: Fourth, our primary cohort included still births. However, although there were 172 cases of stillbirth in the control pool, cases of cleft L/P were all live-born. As a result, the effect of live birth bias could not be evaluated.”

Line 222-224: “A second explanation is that very early fetal losses (e.g., before perception of pregnancy) were not included in the JECS dataset.”

(To the Reviewer: Please also note that this possibility was discussed, but was unlikely as mentioned in the text thereafter.)

Comment 1-5

Statistical problem

Comment 1-5-1

Mean metal level, median metal level and Median with interquartile range are both used in the study, due to the distribution of metal concentration, the expression needs to be uniformed.

Response to Comment 1-5-1

Throughout the manuscript, we uniformly used the median with IQR in this revision. 

Comment 1-5-2

Line 165-166: We found that metal concentrations were heavily skewed owing to outliers. How to tackle with “outliers”?

Response to Comment 1-5-2

We addressed this issue before submission by checking that the primary results were unchanged even when data were log-transformed (and also confirmed that the data were graphically normally distributed after transformation). In addition, following the Editor’s suggestion, we present the results calculated from the log-transformed metal concentrations in this revision (S3 Table). We thank the Reviewer for this comment. 

Comment 1-5-3

The control variables were not enough in the multivariate analysis, which only sex and other metals controlled, how about other risk factors related to CL/P?

Response to Comment 1-5-3

Other maternal risk factors for CL/P (e.g., smoking status or folic acid intake) were used for matching variables. After matching, these variables were well-balanced between cases and controls; thus, we did not perform further adjustment using these variables. Furthermore, in sensitivity analysis 2, we modeled the analysis by incorporating 10 additional maternal factors used for matching, and the results were similar to those of the primary matched analysis. For these reasons, we do not think that the control variables were insufficient. We thank the Reviewer for this comment.

Comment 1-5-4

As there are several metals in the analysis, proper statistical analysis may need, for example, BKMR analysis may be considered.

Response to Comment 1-5-4

We thank the Reviewer for insightful suggestion. We conducted Bayesian kernel machine regression for matched pairs, using the bkmr package in R, and found that the results were similar to those of the primary analysis (i.e., no association).

(Z1, Hg; Z2, Pb; Z3, CD; Z4, Mn)

Although Bayesian kernel machine regression is a sophisticated approach, Journal readers who are not statisticians may not be familiar with it. Thus, we decided to retain logistic regression analysis to present the primary results.

Comment 1-5-5

It’s hard to understand the notes under S1 Table: Major congenital anomalies recorded in Dr0m and Dr1m1 (1: Cleft lip, cleft palate, and cleft lip and palate are listed, bur are omitted from this table.)

Response to Comment 1-5-5

We apologize for the typo and the confused writing in the footnote. We have revised this as “Cleft lip, cleft palate, and cleft lip and palate are also available, but are not listed here for simplicity.” 

Reviewer #2

Comment 2-1

The introduction needs further elaboration about the burden of metal exposures worldwide. Further, the justification of including manganese and selenium is not sufficient. The authors should specify the health impact of metals on the outcome of interest, including the risk estimates.

Response to Comment 2-1

We thank the Reviewer for this important point; the Editor and another Reviewer commented similarly. To explain the rationale for this study, we added a brief description of the hazards of ingesting metals.

As for manganese and selenium, we treated manganese as a heavy metal but omitted selenium from this revision; please refer to our detailed response to comment E3.

Comment 2-2

It is not clear why other cases of congenital anomalies were not considered in the analysis.

Response to Comment 2-2

In the terminology of congenital malformation, syndrome (or syndromic) cases are commonly defined as a patient with malformation of more than one developmental field or body site, with or without identifiable causes such as chromosomal abnormalities (according to Ref. #7). Cleft lip/palate is classified as syndromic or non-syndromic (i.e., without malformation of other body sites) and it is assumed that these two categories have different etiologies. For example, children with Down syndrome (trisomy 21) are at higher risk of cleft lip/palate together with congenital heart defects or gastrointestinal malformation, which constellations are thought to originate from a chromosomal abnormality (although this has not yet been confirmed). We therefore enrolled cases without other malformations in the primary analysis because they were likely to represent non-syndromic cases. 

 However, as mentioned in the text, the classification of syndromic vs. non-syndromic cases may differ between researchers, and it is possible that the environmental factors (heavy metals in our case) may increase the risk of cleft lip/palate conjoint with other potential risk factors in syndromic cases. To account for this, we conducted a sensitivity analysis that involved cases/controls with other anomalies (sensitivity analysis 3), and found that the results were unaffected by case definition. We thank the Reviewer for this comment.

Comment 2-3

Did the author graphically observe the associations by scatter plots (with Loess)? Linear associations are not always observed between toxic exposures and health outcomes. Whether there is any effect in the low exposure levels should be checked graphically. Furthermore, the authors should also explore the associations by grouping the exposures into tertile/quartile.

Response to Comment 2-3

We thank the Reviewer for this thoughtful comment. Another Reviewer suggested an alternative analytical approach, Bayesian kernel machine regression (Biostatistics. 2015 Jul;16(3):493–508). The advantage of this approach is that it can estimate the health effects of multi-pollutant mixtures (e.g., interactions) and investigate exposure-response functions even when the relationship is non-linear. We found that, by graphical inspection, a dose-dependency between metal exposure level and risk of cleft L/P was unlikely. Please refer to the response to Comment 1-5-4. 

Comment 2-4

One of the weaknesses of the study is that exposure at very early gestation is not available. This weakness should be discussed. Further, are there any other unmeasured metals (such as arsenic) that may influence the observed association?

Response to Comment 2-4

We thank the Reviewer for this critical comment. First, we agree with the Reviewer that the lack of data on exposure at very early gestation is the major limitation. We have added the following text as the first limitation:

“However, fetal development of lip and palate begins as early as 4–5 weeks of gestation (2), the period around or before pregnancy perception in most women. Blood sample collection prior to this period is a challenge in a large-scale cohort; it should be acknowledged that there is often a trade-off between research feasibility and scientific basis.”

 With respect to other metals such as arsenic, data would be updated in the future as mentioned in the response to Comment E3.

Comment 2-5

It is better if the authors add a separate table for exposures, including the mean, minimum and maximum values.

Response to Comment 2-5

We thank the Reviewer for this comment. As another Reviewer recommended using either the mean or median uniformly, your recommendation was incorporated into the supplemental table (S4 Table). 

Comment 2-6

It is better for a reader if the authors add paragraphs on the strengths and limitations of the study.

Response to Comment 2-6

We have created a paragraph on the strengths and limitations of the study, as suggested.

Comment 2-7

Minor issues: Page 23, Line 220: missing full-stop; Page 15, Table 1: replace ‘media’ with ‘median’ at the end.

Response to Comment 2-7

We have accordingly corrected these errors. We thank the Reviewer for pointing them out.

---

## [Decision Letter · Decision Letter 1]

27 Jan 2022

PONE-D-21-29327R1Association of maternal heavy metal exposure during pregnancy with isolated cleft lip and palate in offspring: Japan Environment and Children’s Study (JECS) cohort studyPLOS ONE

Dear Dr. Takeuchi,

Thank you for submitting your manuscript to PLOS ONE. After careful consideration, we feel that it has merit but does not fully meet PLOS ONE’s publication criteria as it currently stands. Therefore, we invite you to submit a revised version of the manuscript that addresses the points raised during the review process.

We look forward to receiving your revised manuscript.

Kind regards,

Forough Mortazavi

Academic Editor

PLOS ONE

Journal Requirements:

Additional Editor Comments:

Dear authors,

Thank you for submitting your manuscript to PLOS ONE. The manuscript has been revised according to the comments except for a few points which should be addressed. Please kindly check the manuscript to ensure that all the RECORD checklist items are complied with.

Reviewers' comments:

Reviewer's Responses to Questions

**Comments to the Author**

1. If the authors have adequately addressed your comments raised in a previous round of review and you feel that this manuscript is now acceptable for publication, you may indicate that here to bypass the “Comments to the Author” section, enter your conflict of interest statement in the “Confidential to Editor” section, and submit your "Accept" recommendation.

Reviewer #1: All comments have been addressed

Reviewer #2: All comments have been addressed

2. Is the manuscript technically sound, and do the data support the conclusions?

Reviewer #1: Yes

Reviewer #2: Yes

3. Has the statistical analysis been performed appropriately and rigorously? 

Reviewer #1: Yes

Reviewer #2: Yes

4. Have the authors made all data underlying the findings in their manuscript fully available?

Reviewer #1: No

Reviewer #2: Yes

5. Is the manuscript presented in an intelligible fashion and written in standard English?

Reviewer #1: Yes

Reviewer #2: Yes

6. Review Comments to the Author

Reviewer #1: Abstract needs to be revised, especially the part of results, results of conditional logistic regression which test the association between heavy metal exposure and isolated cleft L/P needs to be demonstrated。

Reviewer #2: The authors addressed all the issues raised by the reviewer. It is better to attach the revision indicating the changes made. Furthermore, the authors also should add the line and page number of the new texts to make the review easy.

7. PLOS authors have the option to publish the peer review history of their article (what does this mean?). If published, this will include your full peer review and any attached files.

Reviewer #1: **Yes: **Jufen Liu

Reviewer #2: **Yes: **Anisur Rahman

---

## [Author Response · Author response to Decision Letter 1]

8 Feb 2022

Reviewer #1

Comment 1-1

Abstract needs to be revised, especially the part of results, results of conditional logistic regression which test the association between heavy metal exposure and isolated cleft L/P needs to be demonstrated.

Response 1-1

Following this suggestion, we have added the results of conditional logistic regression analysis to the Abstract (ln. # 56-58).

“the adjusted odds ratios (per 1 μg/L increase) with 95% intervals were 0.96 (0.91-1.03), 1.01 (0.94-1.08), 1.00 (0.61-1.63) and 1.00 (0.97-1.03) for Hg, Pb, Cd and Mn, respectively”

Reviewer #2

Comment 2-1

It is better to attach the revision indicating the changes made. Furthermore, the authors also should add the line and page number of the new texts to make the review easy.

Response 2-1

We apologize for the inconvenience in the previous version. In this revision, we indicated the changes with the line number.

---

## [Decision Letter · Decision Letter 2]

7 Mar 2022

Association of maternal heavy metal exposure during pregnancy with isolated cleft lip and palate in offspring: Japan Environment and Children’s Study (JECS) cohort study

PONE-D-21-29327R2

Dear Dr. Takeuchi,

We’re pleased to inform you that your manuscript has been judged scientifically suitable for publication and will be formally accepted for publication once it meets all outstanding technical requirements.

Kind regards,

Forough Mortazavi

Academic Editor

PLOS ONE

Additional Editor Comments (optional):

Reviewers' comments:

Reviewer's Responses to Questions

**Comments to the Author**

1. If the authors have adequately addressed your comments raised in a previous round of review and you feel that this manuscript is now acceptable for publication, you may indicate that here to bypass the “Comments to the Author” section, enter your conflict of interest statement in the “Confidential to Editor” section, and submit your "Accept" recommendation.

Reviewer #2: All comments have been addressed

2. Is the manuscript technically sound, and do the data support the conclusions?

Reviewer #2: Yes

3. Has the statistical analysis been performed appropriately and rigorously? 

Reviewer #2: Yes

4. Have the authors made all data underlying the findings in their manuscript fully available?

Reviewer #2: Yes

5. Is the manuscript presented in an intelligible fashion and written in standard English?

Reviewer #2: Yes

6. Review Comments to the Author

Reviewer #2: (No Response)

7. PLOS authors have the option to publish the peer review history of their article (what does this mean?). If published, this will include your full peer review and any attached files.

Reviewer #2: **Yes: **Anisur Rahman

---

## [Editor Report · Acceptance letter]

10 Mar 2022

PONE-D-21-29327R2 

Association of maternal heavy metal exposure during pregnancy with isolated cleft lip and palate in offspring: Japan Environment and Children’s Study (JECS) cohort study 

Dear Dr. Takeuchi:

I'm pleased to inform you that your manuscript has been deemed suitable for publication in PLOS ONE. Congratulations! Your manuscript is now with our production department. 

Kind regards, 

on behalf of

Dr. Forough Mortazavi 

Academic Editor

PLOS ONE